# Air Pollution Increases the Incidence of Upper Respiratory Tract Symptoms among Polish Children

**DOI:** 10.3390/jcm10102150

**Published:** 2021-05-16

**Authors:** Aleksandra Ratajczak, Artur Badyda, Piotr Oskar Czechowski, Adam Czarnecki, Michał Dubrawski, Wojciech Feleszko

**Affiliations:** 1Department of Pediatric Respiratory Diseases and Allergy, Medical University of Warsaw, Żwirki i Wigury 63A, 02-091 Warsaw, Poland; aleksandra.ratajczak@wum.edu.pl; 2Department of Informatics and Environment Quality Research, Faculty of Building Services, Hydro- and Environmental Engineering, Warsaw University of Technology, 20 Nowowiejska Street, 00-653 Warsaw, Poland; artur.badyda@pw.edu.pl; 3Department of Quantitative Methods and Environmental Management, Faculty of Entrepreneurship and Quality Science, Gdynia Maritime University, 83 Morska Street, 81-225 Gdynia, Poland; p.o.czechowski@wpit.umg.edu.pl; 4ARC Rynek i Opinia, Market Research Institute, Słowackiego 12, 01-627 Warsaw, Poland; adam.czarnecki@arc.com.pl (A.C.); michal.dubrawski@arc.com.pl (M.D.)

**Keywords:** particulate matter, respiratory infections, upper respiratory tract symptoms, air pollution, children

## Abstract

A substantial proportion of airway disease’s global burden is attributable to exposure to air pollution. This study aimed to investigate the association between air pollution, assessed as concentrations of particulate matter PM2.5 and PM10 on the upper respiratory tract symptoms (URTS) in children. A nation-wide, questionnaire-based study was conducted in Poland in winter 2018/2019 in a population of 1475 children, comparing URTS throughout the study period with publicly available data on airborne particulate matter. A general regression model was used to evaluate the lag effects between daily changes in PM10 and PM2.5 and the number of children reporting URTS and their severity. PM10 and PM2.5 in the single-pollutant models had significant effects on the number of children reporting URTS. The prevalence of URTS: “runny nose”, “sneezing” and “cough” was positively associated with 12-week mean PM2.5 and PM10 concentrations. In the locations with the highest average concentration of PM, the symptoms of runny nose, cough and sneezing were increased by 10%, 9% and 11%, respectively, compared to the cities with the lowest PM concentrations. This study showed that moderate-term exposure (12 week observation period) to air pollution was associated with an increased risk of URTS among children aged 3–12 years in Poland. These findings may influence public debate and future policy at the national and international levels to improve air quality in cities and improve children’s health.

## 1. Introduction

Air pollution is a complex mixture of different gaseous and particulate components that may cause cardiovascular and respiratory disorders (i.e., asthma, chronic obstructive pulmonary disease) and contribute to increased mortality [1]. Evidence continues to accumulate for pleiotropic detrimental health effects of particulate matter air pollution. Particulate matter (PM) derives predominantly from burning solid fuels for heating or cooking in domestic coal or wood boilers and stoves, manufacturing and extractive industry, agriculture and, to a lesser extent, from burning liquid fuels to power vehicle engines. Particulate matter is a term that encompasses solid and aerosolized liquid particles suspended in the air, described by size distribution, the most common of which are two fractions: PM10 (particulate matter with aerodynamic diameter up to 10 μm) and PM2.5 (with aerodynamic diameter of up to 2.5 μm). Less frequently distinguished are submicron (PM1) and ultrafine (PM0.1) fractions with a diameter up to 1 μm and 0.1 μm, respectively [2].

The nasal mucociliary system is the first line of defense of the upper airways and it is responsible for the clearance of inhaled particles, including harmful particulate matters (PM). The inflammatory response induced by inhalation of air pollutants is linked with up regulation of various pro-inflammatory cytokines and chemokines which could affect the ciliary beating and cause mucus retention and upper respiratory tract symptoms [3]. Respiratory tract infections are essential contributors to morbidity and mortality in childhood [4]. The lack of a fully developed pulmonary, metabolic capacity in children, high respiratory rate and leaky epithelium makes them more susceptible to air pollutants than adults [5]. In children, the small airway caliber allows for a higher chance of being affected by inflammation resulting from air pollution [6,7]. In the last decade, epidemiological studies have shown an association between air pollution and increased incidences of respiratory symptoms and infections, increased hospitalizations for respiratory diseases and premature mortality [8]. 

Whereas evidence of adverse effects of air pollution on lower respiratory tract illnesses in children is increasing, little is known about the effects of air pollution on the incidence of upper respiratory tract symptoms (URTS). The existing epidemiological studies have demonstrated that a short-term increase in outdoor air pollutants, such as PM10, PM2.5, NO_2_, CO and O_3_, is associated with an acute increase in the number of children attending hospitals as outpatients with upper respiratory tract infections [9,10,11]. Studies on the adverse effects of air pollution on upper respiratory tract symptoms (URTS) in children are relatively scarce in Poland and Central Europe [12,13,14]. However, the problem is real: in 2020, the European Environmental Agency reported that Poland occupied the second highest position among European Union countries regarding annual mean concentrations of PM10 and the first one in the case of PM2.5. It was also estimated that the exposure to PM2.5 in Poland contributes to 46,300 premature deaths annually [15].

This study aimed to: (i) evaluate the short-term effects of daily mean concentrations of PM10 and PM2.5 (one to six days after PM concentrations reading) and to (ii) assess the medium-term (12 weeks observation period) effects of daily mean concentrations of PM10 and PM2.5 on pediatric patients aged 3–12 years with URTS in Poland. This study’s results may provide evidence for health risk assessment and the development of public health policies.

## 2. Materials and Methods

### 2.1. Study Area and Population

The study population was representative of the target population. Measurements were performed in 255 representative locations in which daily monitoring of air pollution was made by either State Environmental Monitoring (SEM) stations or an independent commercial monitoring agency (Airly, www.airly.com (accessed on 4 March 2019)) (Figure 1).

The selected locations were diversified due to the size of the location and the region in Poland. 

The locations were then divided into three categories depending on the concentration of PM2.5 and PM10: LOW: Locations where mean PM2.5/PM10 concentrations were low (22.04–26.70 µg/m^3^ for PM2.5 and 27.37–35.37 µg/m^3^ for PM10);MODERATE: Locations where mean PM2,5/PM 10 concentrations were moderate (33.15–37.63 µg/m^3^ for PM2.5 and 47.42–55.41 µg/m^3^ for PM10);HIGH: Locations where mean PM2.5/PM10 concentrations were high (41.95–51.38 µg/m^3^ for PM2.5 and 65.69–88.84 µg/m^3^ for PM10).

The above classification was the basis for conducting data analyses. We checked whether children, in particular subgroups of cities, significantly differ in URTS.

The quartile analysis was used to isolate the groups:

Locations where the mean concentration of particulate matter was low—I quartile;

Locations where the mean concentration of particulate matter was moderate—II and III quartiles;

Locations where the mean concentration of particulate matter was high—IV quartile.

### 2.2. Data Collection

Data on respiratory morbidity were collected using a self-reported questionnaire based on the polish version of the Wisconsin Upper Respiratory Symptom Survey for kids (WURSS-k) [16]. The research tool for collecting data on children’s health was a self-filling questionnaire carried out with the CAWI (computer-assisted web interviewing) technique. The questionnaire was transformed into a smartphone app, which was then downloaded by participants. Every day, at the same time, the respondents received a reminder to fill in the questionnaire (via a dedicated mobile application). Parents of children aged 3–12 years were identified. They assessed their children’s health condition and the occurrence of the URT symptoms in their children every day at approximately 7–8 p.m. Each parent reported their children’s health condition for at least three consecutive days, while those parents who wanted to participate for a period longer than three days were able to do so. In the following weeks, successive parents reported the health condition of their children. The total delivery time was 12 weeks (12 weekly installs), and the measurements were carried out daily. Data were collected between 19 November 2018 and 3 March 2019 (peak air-pollution period and heating season in the entirety of Poland). The parents of the participating children were recruited from an online public opinion research panel—epanel.pl. The sample was selected on a quota basis. Quota sampling is a non-probability sampling method in which researchers create a sample involving individuals that represent a population. Researchers choose these individuals according to specific traits or qualities. These samples can be generalized to the entire population. 

### 2.3. Exposure Assessment/Ambient Air Monitoring

Daily mean concentrations of particulate matter (PM2.5 and PM10) were retrieved from a public database of measurements performed at SEM stations or an independent commercial monitoring agency between 19 November 2018 and 3 March 2019 (Figure 1). For assessing the PM exposure, monitoring stations were identified based on the parent’s smartphone’s geolocation or patients’ addresses of residence), so that each child involved in this study was paired with a mean daily PM concentration. Due to the insignificant number of sensors located outside the cities, the sample did not include children living in the countryside.

We have examined two different time scales—12 week average and short-term exposure (1 day to past six days).

Air pollution data were categorized into quartiles. Evaluations of the effect of high air pollution on the prevalence of URTS were made between the upper, medium and lower quartiles for each pollutant separately.

### 2.4. Questionnaires

Each participant had to fulfill two types of the questionnaire: a preliminary demographic survey at the onset of the study (Appendix A) and subsequently, a daily WURSS-k-based questionnaire (Appendix A). 

The preliminary survey consisted of questions considering sociodemographic characteristics, known allergies, previous respiratory disorders, and various environmental risk factors—child’s home, heating system, type of building, education and smoking habits of its residents. Patients and parents eligible for the study completed a questionnaire on the URTS of their children in a daily mobile diary. For the questionnaire about the children’s respiratory symptoms, the core questions from the standardized questionnaire Wisconsin Upper Respiratory Symptom Survey for Kids (WURSS-k) were used, extended by the addition of questions about eye irritation and headaches. The standardized WURSS-k questionnaire has previously been translated into Polish and validated [16]. An essential part of this questionnaire was a self-assessment of the participating children’s symptoms (cough, runny nose, sneezing, eye irritation) on a Likert scale. Completion of the questionnaire was checked every day by the research team. 

The data collection was blind: the participants were not aware of the purpose of the study. Thus, the potential information bias caused by the knowledge of children’s exposure status was eliminated.

### 2.5. Statistical Analysis

Health effects of ambient air pollutants exposure were estimated by multivariate logistic regression and multivariate linear mixed-effects models

A Generalized Regression Models (GRM) were used to evaluate the lag effects (including lag0 to lag6) between daily changes in particulate matter (PM10 and PM2.5) concentrations and the number of children reporting URTS and the URTS’ severity.

GRMs (Generalized Regression Models) are an extension of the GLM (Generalized Linear Model) family. A general linear model may be treated as an extension of multiple linear regressions for a single dependent variable. GRM models contain, apart from the factors taken into account, the chance factor in case of unknown “confounders”. The chance factor must meet the assumptions (stationarity, normality and does not interfere with the possibility of finding a mathematical solution to the model).

The basic multiple regression model in its general form is as follows:
Y = b_0 + b_1 X_1 + b_2 X_2 + … + b_k X_k,(1)
where: Y—explained variable;b—intercept;X—explanatory variable;k—number of predictors.

GRM is a kind of estimation path that consolidates regression methods and models, prepared for estimating variables in all measurement scales and interactions between endogenous variables. It allows the identification of a cause-and-effect relationship regardless of the measurement scale of independent variables. 

The first step in model quality assessment is to verify how well empirical data fit the model or to test the goodness of fit, with the available error measures applied. The most commonly used metric for goodness fit assessment is the determination co-efficient: R^2 = 1 − (n − 1)/(n − k − 1) (1 − (∑_(t = 1)^n▒(x^_t − x^−^)^2)/(∑_(t = 1)^n▒(x_t − x^−^)^2)),(2)
where:xt—values of variable X at time or period t;x^_t—theoretical value of variable X at time or period t;x^−^—mean value of variable X in a time series on n observations;n—number of observations;k—number of explanatory variables.

This metric shows the goodness of the model’s fit with the empirical data. Its primary advantage is normalization. In this study, this metric also plays an explanatory role. It specifies what portion of the information pool on disease incidence variability could be accounted for by the model and, therefore, the extent to which the identified predictors account for total disease incidence variability.

Eventually, the formula proposed by Makridakis as an extension of the previous variants was adopted as the key selection criterion:
AICC = n(1 + log *fo* (2π)) + n∙lnσ^_k^2 + 2k(3)

### 2.6. Sample Selection

We recruited parents of children aged 3–12 with specific exposure to a potentially pathogenic agent (clinical sample), i.e., particulate matter from an online research panel—epanel.pl. Epanel is one of the largest public opinion polling forums in Poland, owned by the Polish research company ARC Rynek i Opinia, Market Research Institute which has been conducting marketing research on the Polish market for over 25 years. We recruited purposefully on our internet public research panel, directing the recruitment survey to people who lived in the cities where the sensors were located and who belonged to groups with individual historical mean particulate matter concentrations.

The sample of respondents was constructed based on three assumptions:The respondents were to live in air-quality monitored areas in the vicinity of an SEM air quality monitoring station or commercial monitoring agency sensors;The respondents were to live in localities with different historical levels of particulate matter based on measurement results from State Environmental Monitoring or Airly, which allowed to identify the locations with low, moderate or high PM concentrations. Due to the relatively low number of stations located outside the cities, the sample did not include children living in the countryside;The sample covers the country’s entire area and includes various regions of Poland and different sizes of the localities.

## 3. Results

### 3.1. This Ambient Air Monitoring

We found a robust geographical variability in the mean PM2.5 and PM10 concentrations during the study period in particular investigation sites. In northern and north-western regions of Poland (Figure 1A), the PM concentrations were rather low and varied between 22.04 µg/m^3^ 26.70 µg/m^3^ in the case of PM2.5 and between 27.37 µg/m^3^ and 35.37 µg/m^3^ in the case of PM10. In Poland’s central and east-central part, the situations were worse, with the concentrations of PM2.5 in the range of 33.15–37.63 µg/m^3^ and of PM10 in the range of 47.42 µg/m^3^. In the case of the southern and south-east part of the country, however, where the air pollution problem usually occurs most often and with the greatest intensity, the concentrations of PM2.5 varied between 41.95 µg/m^3^ and 51.38 µg/m^3^ and the concentrations of PM10 between 65.69 µg/m^3^ and 88.84 µg/m^3^. 

### 3.2. General Characteristics of the Study Population

The presence of self-reported URTS in the observation period (12 weeks) was frequent: 68% of children reported at least one of the investigated respiratory symptoms. These data suggest that more than half of the population has developed infection symptoms attributed to air pollution in this group of children. The most common URTS, regardless of daily mean PM concentrations, were: runny nose (61%), sneezing (56%), stuffy nose (54%), fatigue (53%) and cough (52%). Symptoms least often reported by children included vision problems (10%), the feeling of sand under the eyelids (10%) and pain in the ear (10%).

In order to exclude misdiagnosis bias, we asked the participants about known airborne allergies in the preliminary questionnaire (Appendix A) and only 17.2% (255/1475) of them confirmed an allergy. What is more, there was no correlation between the number of children in the household and the prevalence of airborne allergy (Appendix A).

### 3.3. Short Term Effects (0 to 6 Days)

There was no significant correlation between the mean daily concentrations of PM2.5 or PM10 with URTI symptom score (Spearman’s correlation). Subsequently, PM10 and PM2.5 concentrations were analyzed in the two-, three-, four-, five- and six- days-lag. However, there was also no statistically significant correlation observed between PM concentrations and the symptom scores for more than two out of seven days in these cases. 

### 3.4. Medium-Term Effects (12 Weeks)

Observations of 12 weeks mean PM10 concentrations and changes in URTI symptom score demonstrated a significant effect indicating an association between the PM10 concentrations and rhinitis in children. The cumulative ratio of rhinitis episodes was reported in 56%, 61%, and 66% of children in the locations with the low, moderate and high daily mean PM10 concentrations, respectively (Z proportion tests, *p* ≤ 0.05) (Figure 2). Similar effects were noted for the stuffy nose symptom, with 52%, 54% and 56% of children respectively, but this effect was statistically insignificant. 

The URTI symptom score demonstrated a strong association between PM10 concentrations and cough in children. The cumulative ratio of cough was reported in 50%, 50% and 58% of children in the locations with the low, moderate and high daily mean PM10 concentrations, respectively (Z proportion tests, *p* ≤ 0.05) (Figure 3). 

Significant differences between the locations were also observed in the case of sneezing, reported in 49%, 55% and 60% of children exposed to low, moderate and high concentrations of PM10, respectively (Z proportion tests, *p* ≤ 0.05) (Figure 4).

The difference in the occurrence of hoarseness (34% vs. 36%) and sore throat (30% vs. 33%) in children living in cities with low vs. high daily mean PM10 concentrations was statistically insignificant (Chi-Square test). 

We have obtained similar results for URTS caused by PM2.5 mean daily concentrations. We also noted an association between the PM2.5 concentrations and the runny nose in children over the 12 weeks. The cumulative ratio of the runny nose episodes was reported in 57%, 61% and 67% of children living in the locations with low, moderate and high daily mean PM2.5 concentrations, respectively (Z proportion tests, *p* < 0.05) (Figure 5). 

Similar effects were noted for the stuffy nose symptoms (53%, 54% and 57% respectively); however, these findings were statistically insignificant. The URTI symptom score demonstrated a strong relationship between the PM2.5 concentrations and cough in children. The cumulative ratio of cough was reported in 49%, 50% and 58% of children in the locations with the low, moderate and high daily mean PM10 concentrations, respectively (Z proportion tests, *p* ≤ 0.05) (Figure 6). 

Remarkable differences between the locations with low, moderate and high daily mean PM2.5 concentrations were also observed in the case of sneezing reported by 51%, 55% and 62% of children, respectively (Z proportion tests, *p* < 0.05) (Figure 7). 

With the increase in the daily mean concentration of PM2.5, difficulties in cognitive functioning among children were observed. Disturbed cognitive functioning was reported in 26%, 29% and 35% of children living in the conditions of low, moderate and high daily mean PM2.5 concentrations, respectively (Z proportion tests, *p* ≤ 0.05) (Figure 8). Prevalence of outcomes long-term lagged mean of PM2.5 and PM10 have been presented on the summary graph (Figure 9).

Other URTS considered in our study i.e., sore throat, hoarseness, loss of appetite, headache, fatigue, irritability, nervousness, difficulty in thinking, redness of the eyes, a sensation of sand under the eyelids, eye discomfort, vision problems, ear pain and feeling of ear clogging showed no significant association with increased PM10 or PM2.5 concentrations.

## 4. Discussion

### 4.1. Principal Findings

In this study, we showed the association between air pollution and URTS in the pediatric population in Central Europe. We further demonstrated that URTS in children during the heating season in Poland were significantly associated with medium-term exposure to PM concentrations. In the 12-week observation period, increased daily mean concentrations of PM10 and PM2.5 were associated with several URTS in children, including runny nose, sneezing and cough (both for PM10 and PM2.5), as well as impaired cognitive functions (for PM2.5 only). 

### 4.2. Strong Points of the Study

It is the first Polish study on the lagged effects of air pollutants concentrations and their associations with children’s URTS. The analysis has been conducted on a relatively large group of participants, using a reliable, self-reported and validated questionnaire (the Polish version of the WURSS-k). Previous studies mainly focused on respiratory infections in general, including pneumonia or acute bronchitis [17,18,19,20,21], but only a few studies have shown an increase in acute upper respiratory symptoms [13]. Darrow et al. [11] observed a 4% increase in emergency department visits for respiratory infections and an 8% increase in visits for pneumonia in the cold-season with higher air pollution levels. These numbers correspond with a one-digit percent increase in URTS in children exposed to higher particulate matter concentrations in our study. Ultimately, our study shows that increased ambient PM concentrations may contribute significantly to the individual’s predisposition to respiratory infections and their severe course, which remains a severe public health issue in the era of the COVID-19 pandemic [22]. Recently, it has been demonstrated that there is an association between higher mortality rates due to new respiratory virus infection (SARS-CoV-2) observed in some European regions and increased average concentrations of PM10 exceeding a daily limit of 50 µg/m^3^ [23]. Our data are in line with these notions and suggest that exceedance of the daily limit value of PM10 and PM2.5 appear to be a significant predictor of respiratory morbidity with serious consequences.

### 4.3. Potential Confounders and Limitations of the Study

This research is not without limitations. Firstly, air pollution is a complex mixture of solid particles, liquids and gases, making it impossible to separate the effects of the constituent molecules in exposure models. The use of two components (PM2.5 and PM10) as air pollutant exposure indicators is insufficient [24]. However, the measurement devices used by the commercial monitoring agency did not allow for the measurement of concentrations of other air pollutants, such as NO_2_ or O_3_ and using only information from SEM stations would not allow us to assess the exposure of residents from most medium and small-sized cities included in the study. It should be emphasized that different components can have an additive, synergistic or even antagonistic effect. Concentrations of PM10 and PM2.5 in the air during the study period could not represent individual exposures. The investigation area was large, country-wide, and other differences in regional characteristics such as climate and lifestyle habits were introduced. Secondly, upper respiratory tract infection symptoms, (e.g., running nose, sneezing and feeling tired) may overlap with the symptomatology of allergic rhinitis and could have been misreported in our self-assessed questionnaire, which is another limitation. Nevertheless, the WURSS-k questionnaire was validated for such cases, also in our preliminary study [16]. Thirdly, although greater overcrowding and poorer housing quality might pose an increased risk of allergy symptoms and coincide with URTS caused by air pollution becoming a limitation of this study, Samoliński et al. demonstrated that, in Poland, impoverished housing standards do not indicate higher risk of sensitization and house dust mite allergy [25]. Considering that many of the recruited participants were attracted to fill out questionnaires for a modest profit, thus being economically weak, it is impossible to differentiate responders according to their socio-economic status, and this might be considered another limitation of our study. Moreover, latencies (the lag effects) led to more difficulties in clarifying the associations. The predicament of showing dependency due to lag effect was raised by Lin et al. in their study, which investigated the association between outpatient visits of acute upper respiratory infections for children and temperature, air pollutants and circulating respiratory viruses in Taiwan [9]. In this study, the authors observed apparent lag effects varying with covariates and their levels throughout evaluating periods, therefore suggesting that using lag models to assess the cumulative relative risks of atmospheric environment and circulating respiratory viruses associated with outpatient visits for acute upper respiratory infections is admissible. Furthermore, the symptoms evaluated at the time of testing were typical symptoms of upper respiratory tract infections, such as running nose, cough or sneezing. These symptoms, however, are non-specific and may be attributed to a variety of causes, including viral infections, toxic effects of air pollution and allergy. Since no further causal investigations, such as blood tests or PCR (polymerase chain reaction) swabs, were performed, we cannot definitely separate infectious from irritative, allergic or other causes. We conducted our research between November and March which is not pollination season in Poland and therefore suggests infection symptoms rather than allergy. Nonetheless, this issue remains a limitation. Another limitation of the present research is that coupling was based on the smartphone’s geolocation, allowing only a rough assessment of adequate exposure. We were concerned about potential exposure misclassification for those cohort members who spent significant amounts of time at daycare centers. In order to conduct a thorough study to determine the relationship between the severity of URTS and the PM10 and PM2.5 dose exposure, children would have to carry a device with themselves that passively collects the sample. Such research would require substantial financial outlays—face-to-face recruitment, high salaries, training of children and parents and highly specialized equipment. However, coupling based on smartphone’s geolocation approach has been recently demonstrated and partly validated in research assessing the effects of traffic-related air pollution on children’s health, calculating the distance to the nearest main road [26,27]. What is more, as reported in most previous ecological studies [28,29], we used the monitoring concentrations to represent ambient air pollution exposures. We were not able to obtain individual exposure to air pollutants, particularly taking into account the size of the examined group. Also, we did not consider the effects of exposure to indoor air pollution, which might affect our findings. Our study’s exposure measurement error was inevitable, which may bias our estimates towards the null [30,31,32]. Despite these limitations, our findings add useful information to understand the association between URTS and air pollutants.

### 4.4. Comparison with Other Studies

Recently, Dyląg et al. have performed a retrospective cross-sectional study, conducted in Cracow in 2018 [13]. The authors observed an increase in acute upper respiratory symptoms and significant positive correlations between weekly mean concentrations of most common air pollutants and viral croup cases per week in the winter months. Notably, upper respiratory tract infections are the major reason for seasonal morbidity in the Northern Hemisphere. Although generally mild and spontaneously solving, they significantly impact the patient’s quality of life and cause relevant problems to the family, society and health care system [33,34], which pose a potentially avoidable factor contributing to the burden of URTS.

Similarly, another recent cross-sectional study has demonstrated a significant increase in respiratory disease hospitalizations, with a typical time lag between the pollutant peak and the event of two to six days [35]. The study was carried out in Poland, and an overall number of over 20 million hospitalizations was captured. The authors of this study, Slama et al., claim that the lag effect, when approached correctly, is the best tool so far to access the impact of the air pollutants in time. Such lag effect for respiratory diseases hospitalizations shows similarities with earlier findings, using the latencies [36,37]. Besides, a study conducted on school children in southern California has also applied the lag effect. In their study, Gilliland et al. have demonstrated that elevated air pollutant levels are associated with increased school absences for respiratory illnesses, peaking five days after increased air pollutant concentrations [38].

The associations between air pollution and childhood upper and lower respiratory tract infections have been observed previously. PM2.5 was demonstrated to increase the risk of infant bronchiolitis [39], viral upper respiratory tract infections and rhinitis in infants [40,41]. Interestingly, experimental studies have demonstrated that the severity of viral respiratory infection in animal and in vitro models can be enhanced by exposure to air pollutants [42,43,44]. Furthermore, evidence suggests that air pollution is associated with lower lung function and decreased lung growth [45]. Several epidemiological studies have documented a positive association between exposure to particulate air pollution and respiratory symptoms, i.e., cough and wheeze, especially among children [46,47]. In this regard, the findings from two Swiss studies showed that reducing exposure to particulate matter PM10 contributes to improved respiratory health, observed through fewer chronic cough cases in children [48]. Similar conclusions were made by the United States Environmental Protection Agency (US EPA). In their study, it has been demonstrated that increased PM caused nose and throat irritation [49]. A study conducted in Switzerland has shown that, adjusted for socio-economic, health-related and indoor factors, declining PM10 was associated in logistic regression models with the declining prevalence of chronic cough, bronchitis, common cold, nocturnal dry cough and conjunctivitis symptoms among children [48]. Another study from Slovakia revealed that cough reflex sensitivity was significantly higher in school-aged girls living in urban areas than school-aged girls living in rural areas [14]. The authors of this research underlined the effect of air pollution on the reported symptom. A study from Taiwan suggested a positive association between outpatient visits for acute upper respiratory infections and ambient environmental factors, including air pollutants [9]. Morgenstern et al. indicated a positive association between traffic-related air pollutants and wheezing, cough without infection, dry cough at night, bronchial asthma, bronchitis and respiratory infections [26]. In this research, significant associations were found between the pollutant PM2.5 and sneezing, runny/stuffed nose during the first year of life. Similar effects were observed for the second year of life.

### 4.5. Conclusions and Further Research

Our findings are consistent with the studies showing improvement of respiratory morbidity in children and declining air pollution levels [50,51]. These findings’ consistency suggests that associations observed in our study between air pollution and respiratory health outcomes may be causal. On the other hand, our results differed from a series of studies on children in the Netherlands. The authors of several Dutch studies reported non-significant associations between exposure to traffic pollution (measured by PM2.5, NO_2_ and soot) and self-reported upper respiratory tract infections [39,52].

## 5. Conclusions

The causality of observed associations between air pollution and URTS in children is still subject to further exploration. Nonetheless, numerous studies have reported adverse effects of exposure to air pollution and children’s respiratory health. Our study shows that moderate-term exposure to air pollution was associated with increased reported URTS among children. We hope that these findings may influence public debate and future policy at national and international levels to improve air quality and improve children’s health. We also anticipate that these findings are of particular interest regarding the current COVID-19 epidemic, in which enhanced mortality was attributed to tobacco smoking and air pollution.

## Figures and Tables

**Figure 1 jcm-10-02150-f001:**
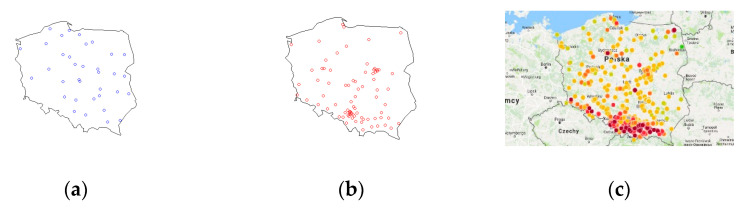
Shows the distribution of the monitoring stations in Poland. (**a**) Public Environmental Monitoring Stations; (**b**) independent private monitoring agency Airly’s stations (http://powietrze.gios.gov.pl/pjp/current (accessed on 4 March 2019); https://airly.org/map/pl/ (accessed on 4 March 2019)); (**c**) distribution of air pollution depending on the region in Poland—PM2.5 concentrations reading. The PM concentrations are illustrated in the form of colors varying from green (good air quality—PM2.5: 0–36 µg/m^3^) through orange (allowable standard—PM2.5: 36–84 µg/m^3^) to red (emission repeatedly exceeded and health hazard—PM2.5: 84 µg/m^3^ and higher).

**Figure 2 jcm-10-02150-f002:**
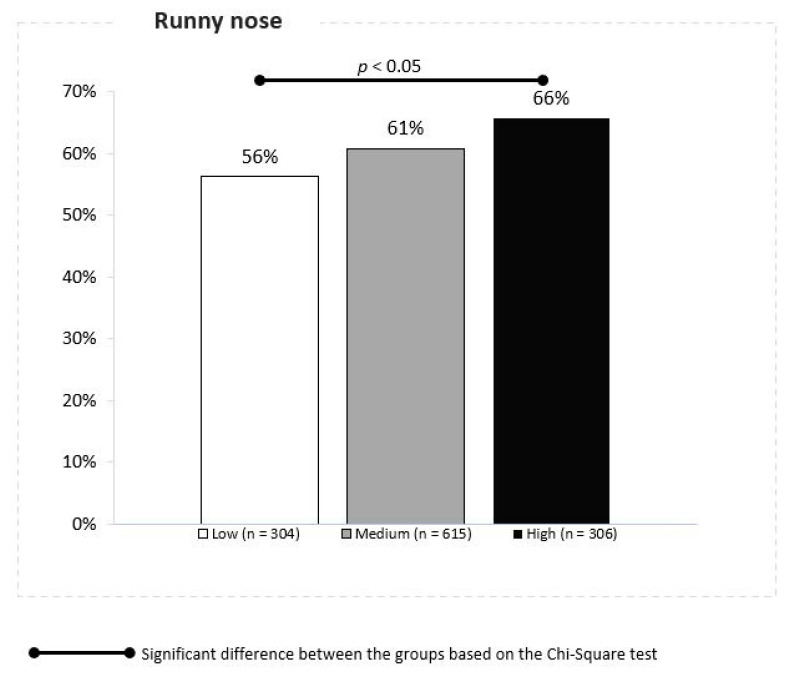
Association between the PM 10 concentrations and runny nose.

**Figure 3 jcm-10-02150-f003:**
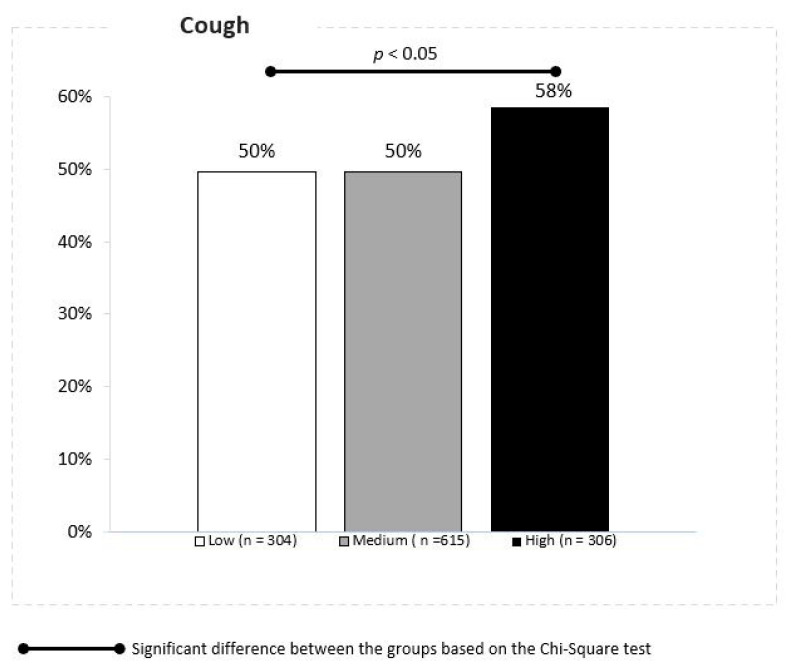
Association between the PM 10 concentrations and cough.

**Figure 4 jcm-10-02150-f004:**
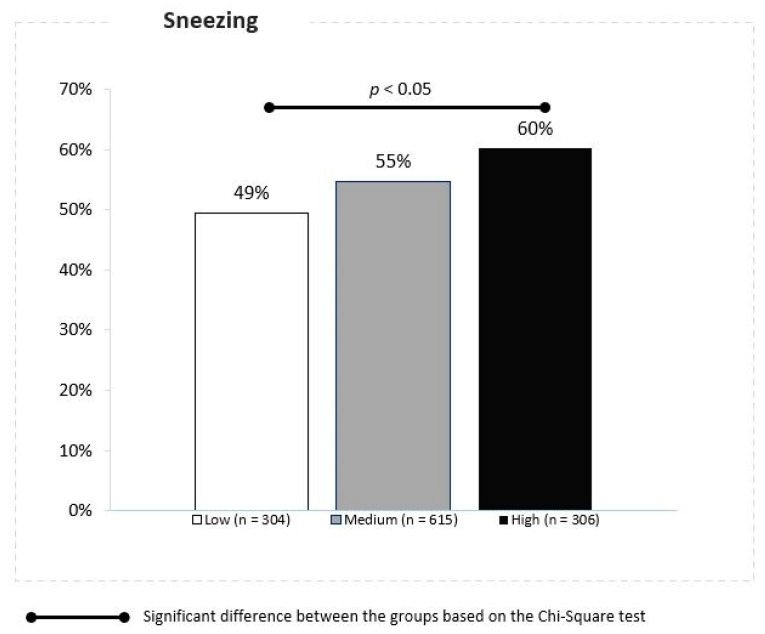
Association between the PM 10 concentrations and sneezing.

**Figure 5 jcm-10-02150-f005:**
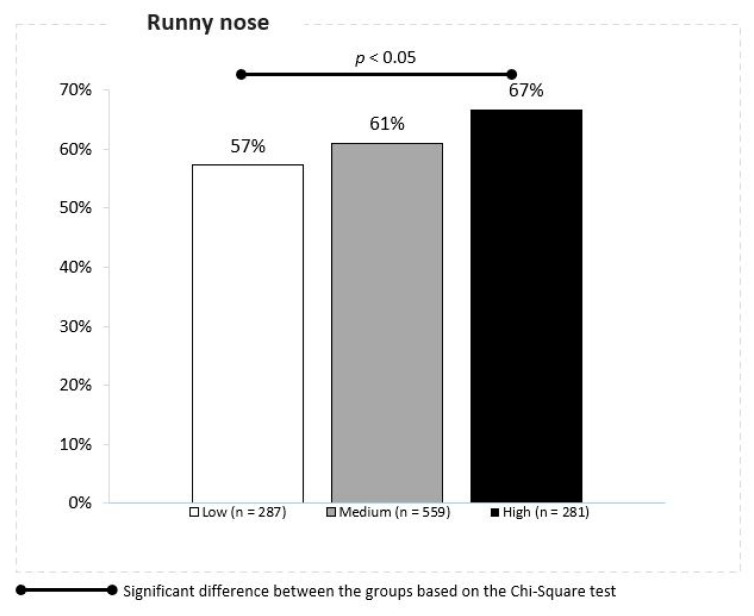
Association between the PM 2.5 concentrations and runny nose.

**Figure 6 jcm-10-02150-f006:**
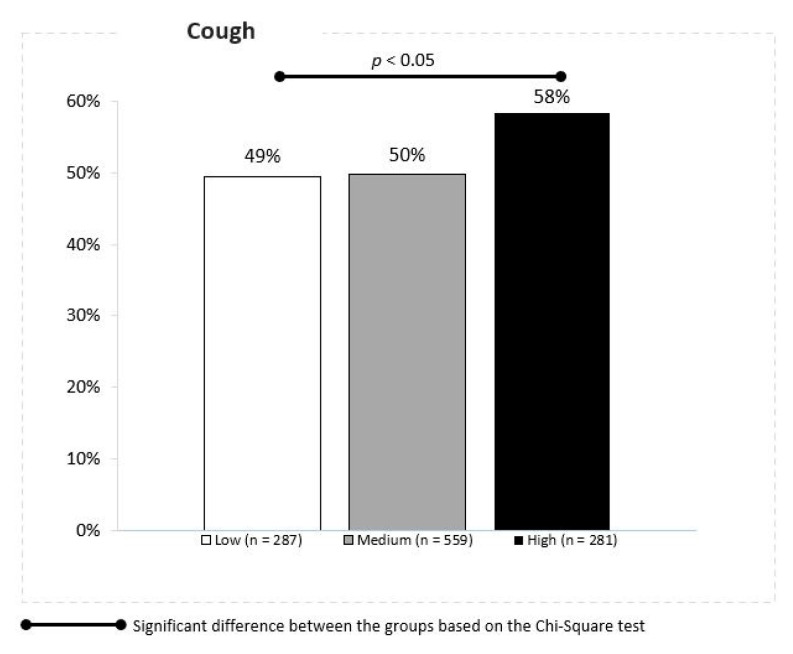
Association between the PM 2.5 concentrations and cough.

**Figure 7 jcm-10-02150-f007:**
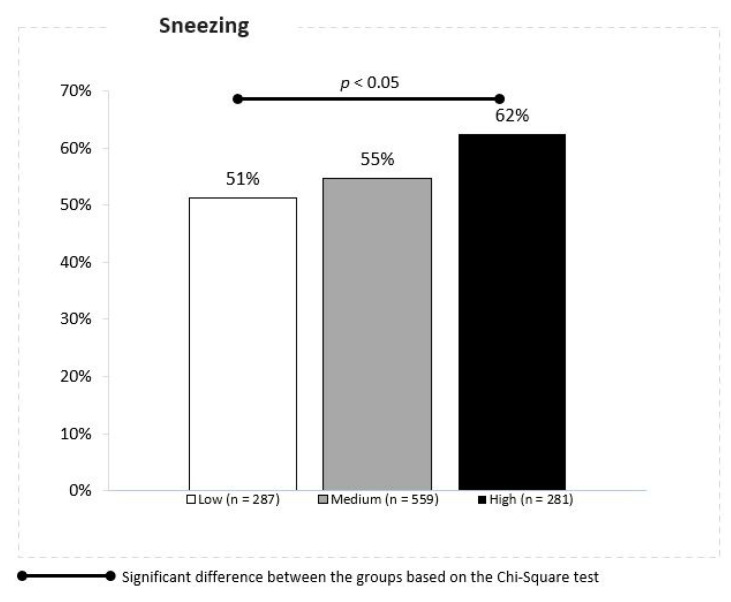
Association between the PM 2.5 concentrations and sneezing.

**Figure 8 jcm-10-02150-f008:**
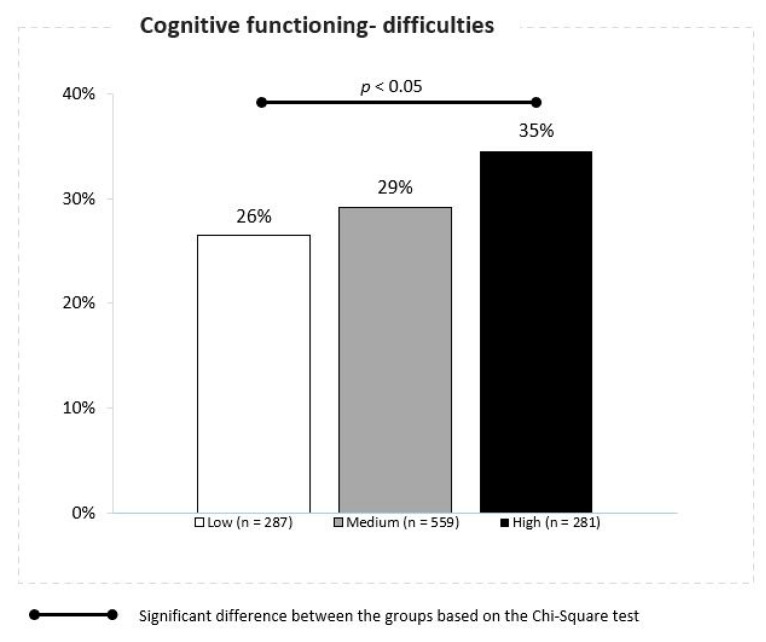
Association between the PM 2.5 concentrations and disturbed cognitive functioning.

**Figure 9 jcm-10-02150-f009:**
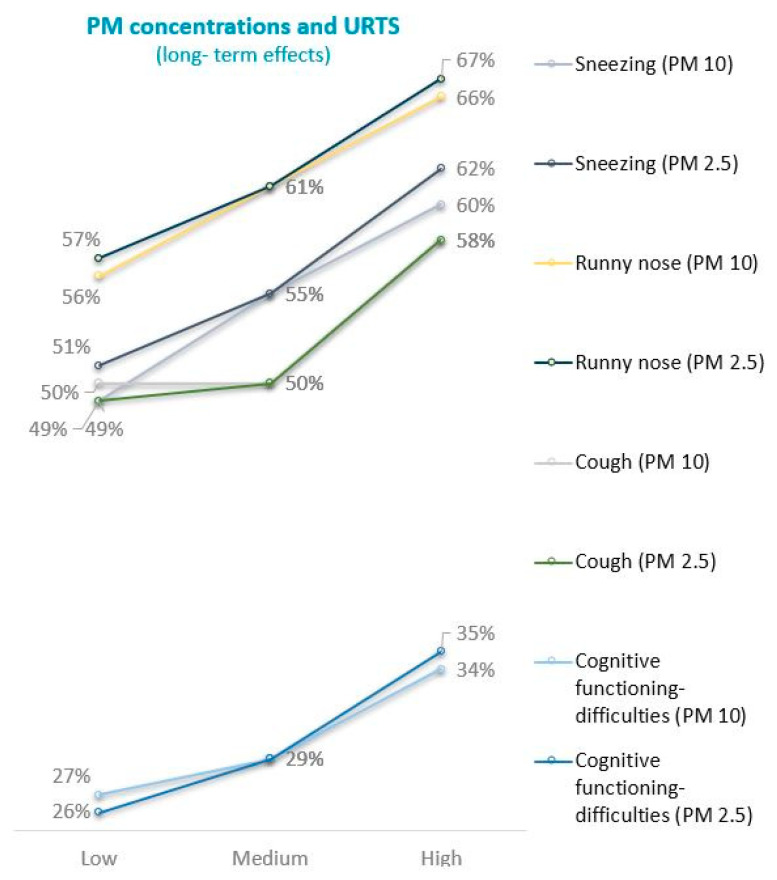
A single time-series figure with prevalence of outcomes long-term lagged mean of PM2.5 and PM10.

## Data Availability

Air quality data available in a publicly accessible repository: State Environmental Monitoring, http://powietrze.gios.gov.pl/pjp/ (accessed on 4 March 2019); Independent private monitoring agency Airly, https://airly.org/map/pl/ (accessed on 4 March 2019)

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
