# Peer review of "Air Pollution Increases the Incidence of Upper Respiratory Tract Symptoms among Polish Children"

_jcm, 2021, doi:10.3390/jcm10102150_

Round 1

Reviewer 1 Report

The researchers investigated the effect of air pollution on risk of URTS among Polish children. The researchers report an association between 12 weeks mean PM exposure and URTS in children. The findings have important public health implications; however, there are also major limitations as described below that needs to be addressed to fully appreciate their findings.

  1. Manuscript issues: A) The introduction is too long and often reads more like one for a review paper than a research paper. It can be more focused. B) This paper does not need 85 references. Making the overall writing more focused can reduce the references by half, at least. C) Table and figures: The importance of Figure 1 is not clear. It would be better to provide short text regarding the variables used in the analysis and the questionnaire (figure 2 &3) can be provided as online supplement. Not sure the role of Table 1 & 2 in the paper as nothing is mentioned about it the text. Figure 4-7 is not much helpful in understanding the findings. A single table with findings for both short- and mid-term exposure.
  2. It was difficult to understand the Methods of the study. It is not clear how the samples were selected. What is epanel.pl and what traits were used to select the sample? Why is the selected sample representative of the study population? What is the time period of the study? The authors mentioned that it was from November to Match. So, do they have ~120 filled questionnaires for each child? Then the study has a panel study pattern for daily outcome that provides the scope for time-series analysis of the data.
  3. A single time-series figure with prevalence of outcomes by date with short- and mid-term lagged mean of Pm2.5 and PM10 would be helpful to appreciate the relationship between those.
  4. The authors mention multivariate Logistic and linear regression (Line 186-7) and then GRM. It is not clear what they have used for what. Did they mean multivariable GRMs? And what are the variables were adjusted for and why?
  5. The discussion is also too long and not focused. It surely can be tightened.

Author Response

Please find the responses in the attached file.

Reviewer 2 Report

This study offers some support to the cause of improving children's health through improved air quality but there are some methodological issues and issues with structure.

The introduction and discussion are overly long and the focus could be improved. It is not clear what is already known about any association between upper airway infection and pollution and why an association might be expected. Is this through inhalation of small pathways and effects upon immunity or a short term effect upon expression of symptoms? Previous work looking at this association should be mentioned first in the introduction  and any discrepencies between this study and previous work discussed critically in the discussion. A clear rationale about why an effect on upper airway infections should be presented in the introduction rather than focussing extensively on lower respiratory tract infections.

Methodologically, greater consideration should be given to whether these symptoms truly represent upper respiratory tract infection or asthma or allergy.

It was not clear whether confounding factors such as socioeconomic status or other geographically patterned factors might explain the association with 12 week averages in the absence of a dose response or lag effect.

Author Response

(The authors gave the same response as above.)

Round 2

Reviewer 2 Report

There introduction now has more structure and provides clearer reasoning for the study.

I remain concerned that cough, sneezing, runny nose etc are variably referred to as upper respiratory tract symptoms URTS (which could have a number of causes including allergy) and upper respiratory tract infection URTI. There are other causes of allergy beyond pollen eg house dust mite and this is where correction for socioeconomic status in my view would be important. Higher pollution might be in a feature of poorer areas with greater overcrowding and poorer housing quality, respectively increasing the risk of URTIs and house dust mite sensitisation and allergy.

It seems a shame to have invested in a complex GRM analysis but not to have considered confounders.

Greater consideration is required regarding why there is no dose dependent or lag effect, this suggests pollution might not be the prime driver of the URTS outcome.

Author Response

(The authors gave the same response as above.)
